# Identification of Novel Biomarkers for Early Diagnosis of Atherosclerosis Using High-Resolution Metabolomics

**DOI:** 10.3390/metabo13111160

**Published:** 2023-11-18

**Authors:** Syed Wasim Sardar, Jeonghun Nam, Tae Eun Kim, Hyunil Kim, Youngja H. Park

**Affiliations:** 1Omics Research Center, Korea University, Sejong 30019, Republic of Korea; syedwasim336@gmail.com (S.W.S.); kh0313k@korea.ac.kr (T.E.K.); kimhyunil7071@gmail.com (H.K.); 2Artificial Intelligence (AI)-Bio Research Center, Incheon Jaeneung University, Incheon 22573, Republic of Korea; jhnam77@gmail.com; 3Metabolomics Laboratory, College of Pharmacy, Korea University, Sejong 30019, Republic of Korea

**Keywords:** atherosclerosis, mass spectrometry, untargeted metabolomics, targeted metabolomics, biomarker

## Abstract

Atherosclerosis (AS) is a metabolic disorder and the pre-stage of several cardiovascular diseases, including myocardial infarction, stroke, and angina pectoris. Early detection of AS can provide the opportunity for effective management and better clinical results, along with the prevention of further progression of the disease. In the current study, an untargeted and targeted metabolomic approach was used to identify possible metabolic signatures that have altered levels in AS patients. A total of 200 serum samples from individuals with AS and normal were analyzed via liquid chromatography–high-resolution mass spectrometry. Univariate and multivariate analysis approaches were used to identify differential metabolites. A group of metabolites associated with bile acids, amino acids, steroid hormones, and purine metabolism were identified that are capable of distinguishing AS-risk sera from normal. Further, the targeted metabolomics approach confirmed that six metabolites, namely taurocholic acid, cholic acid, cortisol, hypoxanthine, trimethylamine N-oxide (TMAO), and isoleucine, were found to be significantly upregulated, while the concentrations of glycoursodeoxycholic acid, glycocholic acid, testosterone, leucine, methionine, phenylalanine, tyrosine, and valine were found to be significantly downregulated in the AS-risk sera. The receiver operating characteristic curves of three metabolites, including cortisol, hypoxanthine, and isoleucine, showed high sensitivity and specificity. Taken together, these findings suggest cortisol, hypoxanthine, and isoleucine as novel biomarkers for the early and non-invasive detection of AS. Thus, this study provides new insights for further investigations into the prevention and management of AS.

## 1. Introduction

Atherosclerosis (AS) is a multifactorial process with global economic and medical importance, with an increased prevalence to cause mortality and morbidity worldwide, and is the main contributor to cardiovascular diseases including stable angina pectoris, myocardial infarction, and stroke [1]. The initiation of AS results from the accumulation of plaque in the sub-endothelial space of arteries, leading to insufficient blood flow and oxygen delivery to vital organs [2,3] It is also characterized as a chronic, multifactorial, and progressive disease showing notable variations in metabolites during different pathological stages [4]. At the initial stage of AS, metabolic imbalances such as a decrease in glucose and glycine and an increase in low-density lipoprotein (LDL) cholesterol may be observed, but without any clear clinical symptoms [5]. As the disease progresses, the narrowing of the arteries limits the blood flow, resulting in symptoms such as hypertension, headaches, and other clinical symptoms. In the meantime, the body may also experience changes in the metabolism of amino acids, phenylalanine, and methionine [4]. In the advanced stage of AS, the plaque becomes unstable and ruptures, leading to thrombosis formation, along with imbalances in the metabolism of sugars, fatty acids, amino acids, choline, and cholesterol [5]. Despite scientific discoveries and significant progress in the treatment of AS, the disease remains a major killer globally [4]. This is due to the lack of effective methods for the detection of AS in its initial stages and a poor understanding of the disease pathophysiology. According to recent epidemiologic statistics released by the World Health Organization and the American Heart Association, cardiovascular diseases stand as the primary reason for death worldwide, and from heart attacks, approximately 24 individuals die every minute in the world [6]. Presently, coronary artery angiography is a method of choice for the diagnosis of AS [7], but its high cost and invasive nature restrict its widespread application in clinical diagnosis or for monitoring the progression of the disease. Generally, coronary artery angiography is only used when clinical and biochemical features strongly suggest the presence of the disease, and its preventive benefits are also limited [8]. Therefore, the discovery of biomarkers for early detection, thus preventing progression to angina pectoris and myocardial infarction accompanied by a high morbidity and mortality rate, is a therapeutic and prognostic intervention.

Currently, high-resolution metabolomics, a comprehensive characterization of metabolites such as organic acids, nucleic acids, amino acids, and lipids in cells, tissues, blood, and biofluids, is getting attention as a novel approach for diagnosing and tracking numerous diseases [9,10]. The identification of small-molecule metabolites and their roles in several biological processes has not only enhanced our perception of the pathophysiology of multiple conditions but has also opened new avenues for the development of novel treatments [11,12]. Techniques involved in metabolomics consist of nuclear magnetic resonance (NMR), gas or liquid chromatography coupled with tandem mass spectrometry (GC/LC-MS/MS) [13]. Advanced metabolomics techniques based on mass spectrometry play a vital role in clinical research, diagnosis and treatment of disease, drug characterization, and environmental and agricultural research [14,15,16]. These novel techniques enable the extraction of information on global metabolic pathways from thousands of metabolites present in biological samples. However, identifying novel biomarkers with diagnostic roles is a clinical challenge for AS.

Therefore, in this study, untargeted and targeted high-resolution metabolomics (HRM) with a stepwise identification workflow was applied to serum samples from normal and AS groups to identify the metabolic signatures for the early detection of AS and provide new insight into the understanding of the pathophysiologic basis of AS in the developmental stage.

## 2. Materials and Methods

### 2.1. Chemicals and Reagents

All analytical standards for target compounds used in the study were of high purity grade (>99%) and were purchased from Sigma-Aldrich (St. Louis, MO, USA). HPLC-grade water was purchased from J.T. Baker (Phillipsburg, NJ, USA) and acetonitrile from Tedia (Fair Lawn, NJ, USA). Individual stock solutions of all target compounds were prepared in water or methanol as per the manufacturer’s instructions and stored at −20 °C before use.

### 2.2. Sample Collection

This project protocol was approved by the Korea University Institutional Review Board (approval no. IRB-2021-0063). Blood samples used in this study were provided by Chungbuk National University Hospital members of the Korea Biobank Network. According to a Korea Biobank Network representative, all participants went through routine health assessments. Additionally, measurements of pulse wave velocity (PWV) and ankle-brachial index (ABI) were performed for the diagnosis of atherosclerosis. Individuals whose PWV was ≥13 m/s and ABI fell within the range of 0.91 to 1.29 were identified as patients with atherosclerosis. To obtain serum, whole blood was centrifuged at 2000× *g* for 10 min to remove the cell pellet, and then the serum was stored at −80 °C until further use. The serum samples were divided into two groups: the normal group (n = 100) and AS (n = 100).

### 2.3. Sample Preparation

Briefly, 50 μL of each serum sample was combined with a 100 μL mixture of acetonitrile and isotope-labeled standards, including [3-methyl-13C]-caffeine, [13C5,15N]-l-methionine, and [dimethyl-D6]-N, N-diethyl-M-toluamide (95:5). The mixture was vortexed for 5 min and centrifuged at 15,000× *g* for 10 min at 4 °C to remove precipitated proteins and extract metabolites. Subsequently, the supernatant was transferred into a 1 mL vial for instrumental analysis.

### 2.4. LC-MS/MS Analysis

Metabolite determination was performed using an Ultra Performance Liquid Chromatography system (Agilent 1290 Infinity Quaternary, Santa Clara, CA, USA) coupled with an Agilent Q-TOF 6550 mass spectrometer. Chromatographic separation was performed using a stationary-phase Hypersil GOLD aQ-C18 column (100 × 2.1 mm, 1.9 μm; Thermo Fisher Scientific, Waltham, MA, USA). Solvent A, consisting of water, and solvent B, Acetonitrile, both acidified with 0.1% formic acid, were used as a mobile phase. The gradient flow was programmed as follows: 0.0–1.0 min, 5% B; 1.0–9.0 min, 45% B; 9.0–12.0 min, 90% B; 12.0–13.5 min, 90% B; and 13.5–13.6 min, 5% B. The flow rate, sample injection volume, and capillary voltage were 0.4 mL/min, 3 μL, and 3.5 kV, respectively. The drying and sheath gas temperatures were both 250 °C. The data were acquired using a mass-to-charge ratio (*m*/*z*) ranging from 50 to 1000, and electrospray ionization (ESI) was performed in positive mode. To eliminate the leftover metabolites between samples, a blank sample consisting of 100% acetonitrile was analyzed among the real samples. For reliability and reproducibility, sample batches were randomized, and each sample was analyzed in triplicate.

### 2.5. Untargeted Metabolite Profiling

To identify the metabolic features that separate the AS patients from normal, multivariate and univariate analyses were carried out. The MSConvert 3.0 (Proteowizard, Palo Alto, CA, USA) was first used to convert the data files from ‘raw data’ to ‘mzXML’ file format. The converted mzXML files were analyzed using apLCMS 6.3.8 software to gather information such as the *m*/*z* value, retention time, and ion intensity of each compound [17]. The files obtained from apLCMS were analyzed further using xMSanalyzer [18] to ensure reproducibility and maximum reduction in the batch effect caused by daily changes in LC-MS conditions. During the xMSanalyzer processing, log_2_ transformation and quantile normalization were used to filter and transform the data. The finally processed data were used in a statistical analysis using xmsPANDA [19]. Investigation of overall metabolic alterations between two groups with xmsPANDA provides the insight to distinguish the group and makes it possible to specify the groups. The significantly differential metabolites with xmsPANDA were further processed with a statistical analysis of the student’s *t*-test (*p* ≤ 0.05) to compare the means between two groups and further multiple testing corrections, Benjamini and Hochberg false discovery rate, and adjusted *p*-values (FDR q ≤ 0.05) to correct for the occurrence of false positives [20]. The selected metabolites were further processed with hierarchical cluster analysis (HCA) and principal component analysis (PCA) using the xmsPANDA package based on limma developed by Emory University. PCA was used to decrease the number of features and dimensions representing the samples into a principal component, while HCA used a bi-dimensional clustering method to visualize the sample features in a heat map based on dissimilarity. The scheme of the experiment was illustrated according to the metabolic signatures of the two groups (Figure 1). Manhattan plots with false discovery rates (FDRs) identified the significantly differential metabolites, whose levels were significantly different between AS and healthy normal samples, according to FDR (q ≤ 0.05). The color-coded dots were depicted in the Manhattan plot by constructing an axis with *m*/*z* as the horizontal axis and –log10(p) as the vertical axis. Significant features (FDR, q ≤ 0.05) from the Manhattan plot were further subjected to PCA and HCA. Differently colored dots in the Manhattan plot were used to represent the separation between groups in PCA and HCA. The significant features identified by the comparisons of both groups using xmsPANDA at the criteria of Limma FDR q ≤ 0.05 were annotated using xMSannotator [21], the Human Metabolome Database (HMDB) (https://hmdb.ca (accessed on 12 October 2023)), and the Kyoto Encyclopedia of Genes and Genomes (KEGG) (https://www.genome.jp/kegg (accessed on 20 October 2023)) databases [22,23]. Limma is an R/Bioconductor software package that provides an integrated solution for analyzing data from gene expression experiments. A mass tolerance of 10 ppm and six adducts [M + H]+, [M + Na]+, [M + K]+, [M + NH_4_]+, [M + H-H_2_O]+, and [M + H-2H_2_O]+ were used to annotate the significant features. In addition, HMDB IDs were obtained to further analyze the pathway analysis in MetaboAnalyst.

### 2.6. Untargeted Pathway Analysis

Pathway analysis aims to analyze the high-throughput data, exploring relevant groups of related metabolites that are altered in AS samples in comparison to a healthy normal. The pathway analysis helps the researcher discover that biological themes and biomolecules are crucial to understanding the disease condition under study. For the identification of potentially altered metabolic pathways in healthy normal versus atherosclerosis patients, the recorded HMDB ID served as input for MetaboAnalyst 5.0 (www.metaboanalyst.ca (accessed on 20 October 2023)).

### 2.7. Targeted Metabolite Profiling of the Significantly Altered in Untargeted Metabolomics

The serum samples from the normal and AS groups were treated using various methods to determine the significantly altered metabolites. Cortisol concentration was determined using the method reported by [24], bile acids were measured using the methodology described by [25], and amino acids and hypoxanthine were analyzed using the methods of [26] and [27], respectively. Tandem mass spectrometry (MS/MS) analysis was conducted using a Triple Quad Mass Spectrometer (Agilent 6490A) with an ESI interface operating in positive mode. The samples were first scanned in the mass range (*m*/*z*) of 50–1000, and then collision energies of 0, 5, 10, 15, and 20 V were applied to get highly abundant fragment ions of the putative metabolites. Chromatographic separation was performed using an Eclipse Plus C18 column (100 × 2.1 mm, 1.8 μm; Agilent USA). The concentrations of identified metabolites in serum samples from healthy normal and AS groups were quantified using eight-point calibration curves. The limit of detection and limit of quantification under the present chromatographic conditions were calculated based on a signal-to-noise ratio of 3 and 10, respectively. Each sample was analyzed in triplicate, and the results were presented as the mean ± standard error of the mean (SEM). The concentrations of targeted metabolites were calculated relative to the peak height of the external standards within the limit of detection (LOD) and quantification (LOQ) ranges. Putative metabolites were analyzed using Prism 7.0 (GraphPad Software, San Diego, CA, USA) to estimate the relative intensities among the two groups.

Furthermore, to validate the clinical effect of these metabolites in the diagnosis of AS, the prediction ability of significant metabolites was assessed with a single-metabolite-based receiver operating characteristic curves (ROC) analysis and a multiple-metabolite-based random forest analysis using MetaboAnalyst. The ROC curve shows the optimal number of variables, sensitivity, specificity, and area under the curve (AUC) of each metabolite that could differentiate between the two groups. AUC is typically the preferred approach for assessing the performance of potential biomarkers; the greater the AUC, the better the prediction of the model [28].

## 3. Results

### 3.1. Characteristics of the Study Population

In total, 200 serum samples from 100 normal and 100 AS individuals were used for metabolomic analysis (Figure 1). Based on the student’s *t*-test, no statistical differences were observed in terms of body mass index, diastolic blood pressure, or total triglyceride among the two groups. Meanwhile, systolic blood pressure, level of fasting blood glucose, and high-density lipoprotein cholesterol were higher in AS patients, while low-density lipoprotein cholesterol and total cholesterol were higher in normal people, but those parameters were in the normal ranges (Table 1). The low levels of low-density lipoprotein cholesterol and total cholesterol, along with the elevated level of high-density lipoprotein cholesterol in AS patients, can be attributed to the administration of medications such as aspirin and clopidogrel, atorvastatin, amlodipine besylate, and bisoprolol.

### 3.2. Determination of Metabolomic Signature in Two Groups

To investigate the metabolomic signature that could discriminate between normal and AS groups, untargeted metabolomics were performed. The raw data extracted from apLCMS containing *m*/*z* values and intensities of 17,336 features were subjected to the xMSanalyzer. With the growth of metabolomics research, more and more studies are conducted on large numbers of samples. Samples often need to be processed in multiple batches or days on the same LC-MS. Across batches or days, we often observe different data characteristics. With batch or day correction of the xMSanalyzer, 4819 features were extracted and input into the xmsPANDA. Multivariate analysis using FDR led to the selection of 1244 significant features (q ≤ 0.05), as shown in Figure 2A. The significant features were further subjected to unsupervised HCA (Figure 2B) and PCA (Figure 2C) analysis. PCA reduces the number of features and dimensions representing samples into a principal component, while HCA provides samples and features through a bi-dimensional clustering method in the form of a heat map based on dissimilarity. The results of HCA and PCA showed a tendency to separate normal from AS. Supervised partial least-squares discriminant analysis (PLS-DA) was performed then since the classification of two groups was not clear using unsupervised methods. PLSDA showed a clear separation between the normal and AS groups (Figure 2D).

### 3.3. Identification of Potential Metabolites between Normal and AS Groups

The significant metabolites between two groups from xmsPANDA with a Manhattan plot (FDR q ≤ 0.05) were determined for novel biomarker discovery. The HMDB (https://hmdb.ca) and xMSannotator [18] were used for the annotation of the significant features obtained from the Manhattan plot with FDR q ≤ 0.05. The HMDB IDs of these metabolites were used to identify the top affected pathways in MetaboAnalyst 5.0. The peak intensities of all the metabolites related to pathways were measured and visualized using bar graphs. Steroid hormone biosynthesis, bile acid biosynthesis, TMO biosynthesis, and amino acid biosynthesis were considered to be responsible for the separation between normal and AS sera (Figure 3). The metabolites involved in selected pathways, L-Valine (*m*/*z*: 156.0421 [M + K]+), L-Arginine (*m*/*z*: 175.1186 [M + H]+), glycocholic acid (*m*/*z*: 466.3196 [M + H]+), hypoxanthine (*m*/*z*: 159.0275 [M + Na]+), cortisol (*m*/*z*: 385.1985 [M + Na]+), testosterone (*m*/*z*: 271.2062 [M + H-H2O]), taurocholic acid (*m*/*z*: 516.2995 [M + H+], glycourodeoxycholic acid, taurodeoxycholic acid (*m*/*z*: 541.3287 [M + ACN + H]), cholic acid (*m*/*z*: 839.5574 [2M + Na]+), taurolithocholic acid (*m*/*z*: 522.2649 [M + K]+), trimethylamine N-oxide (TMAO), glutamic acid (*m*/*z*: 277.1030 [M + H]+), isoleucine (*m*/*z*: 132.1011 [M + H]+), leucine (*m*/*z*: 132.1018 [M + H]+), methionine (*m*/*z*: 150.0582 [M + H]+), phenylalanine (*m*/*z*: 166.0863 [M + H]+), proline (*m*/*z*: 116.0698 [M + H+], and tyrosine (*m*/*z*: 226.0435 [M + 2Na-H]) were dramatically altered in normal and AS groups (Figure 4).

### 3.4. Validation of the Levels of Metabolites with Targeted Metabolomics

To validate the results and identify simplified serum metabolites that would be more practical for identifying AS risk, the concentrations of taurocholic acid, glycoursodeoxycholic acid, glycocholic acid, taurodeoxycholate, cholic acid, taurolithocholate, cortisol, hypoxanthine, testosterone, TMAO, L-arginine, glutamic acid, isoleucine, leucine, methionine, phenylalanine, proline, tyrosine, and valine were determined in normal and AS serum samples using LC-MS/MS (Figure 5). The concentrations of these compounds in both sera samples were calculated using an external standard calibration curve. Based on the results, taurocholic acid, cholic acid, cortisol, hypoxanthine, TMAO, and isoleucine were found to be significantly upregulated in AS samples. Meanwhile, the concentrations of glycoursodeoxycholic acid, glycocholic acid, testosterone, leucine, methionine, phenylalanine, tyrosine, and valine were found to be significantly downregulated.

In addition, the diagnostic potential of significant metabolites was evaluated via the ROC curve and random forest analysis. The ROC curve showed the optimal number of variables, sensitivity, specificity, and AUC of each metabolite that could differentiate between the two groups. Three metabolites, including cortisol, hypoxanthine, and isoleucine, were found to have AUC values of 0.834, 0.935, and 0.792, respectively, with a 95% confidence interval (CI) to distinguish AS patients from normal (Figure 6). Based on the three metabolites (cortisol, hypoxanthine, and isoleucine), a random forest diagnostic model was developed. Remarkably, the model achieved a better AUC of 0.96, surpassing the individual AUC values for each metabolite, as illustrated in Figure 7.

## 4. Discussion

In the present study, untargeted and targeted metabolomic approaches using liquid chromatography–high-resolution mass spectrometry were applied to determine the metabolomic signature that could differentiate between the normal and AS groups and to identify early biomarkers of the AS disease. Using untargeted metabolomics, a group of metabolites, including TMAO, bile acids, amino acids, and steroid hormones, were identified that are capable of distinguishing AS-risk sera from normal. Further, the targeted metabolomics approach confirmed that six metabolites, namely taurocholic acid (AUC-ROC, 0.569), cholic acid (AUC-ROC, 0.625), cortisol (AUC-ROC, 0.834), hypoxanthine (AUC-ROC, 0.935), TMAO (AUC-ROC, 0.663), and isoleucine (AUC-ROC, 0.792), were found to be significantly upregulated, while the concentration of glycoursodeoxycholic acid (AUC-ROC, 0.658), glycocholic acid (AUC-ROC, 0.669), testosterone (AUC-ROC, 0.641), leucine (AUC-ROC, 0.848), methionine (AUC-ROC, 0.870), phenylalanine (AUC-ROC, 0.829), tyrosine (AUC-ROC, 0.818), and valine (AUC-ROC, 0.861) were found to be significantly downregulated in the AS-risk sera. The discovery of these differentially expressed metabolites provides valuable insights into the metabolic alterations associated with AS. Additionally, ROC curves were used in this study to increase the specificity and sensitivity of biomarkers by examining the AUCs in the normal and AS groups. If the AUC-ROC value is greater than 0.7, the metabolite is relatively exclusive and can be considered an early diagnostic biomarker [29]. Three upregulated metabolites, including cortisol, hypoxanthine, and isoleucine, were found to have AUC values of 0.834, 0.935, and 0.792, respectively, with a 95% CI, indicating the potential of these metabolites to serve as early biomarkers of AS. The elevated levels of biomarkers in the blood are preferable for diagnosing, monitoring, or treating a particular disease. Therefore, only the upregulated metabolites were selected in this study.

The results of our study are consistent with the results of previous studies; for instance, previously conducted clinical studies have shown a correlation between elevated levels of TMAO in plasma and AS [30,31]. TMAO has been suggested to potentially exacerbate inflammatory responses in the vascular wall, induce the production of reactive oxygen, and impair cholesterol reverse transport, which plays a role in the development of atherosclerosis [32]. In addition to the increased production of reactive oxygen species, the level of methionine, which is the precursor of cysteine and glutathione, decreased and caused oxidative stress. Similarly, alterations in bile acids and amino acids have been linked to dyslipidemia and oxidative stress, leading to AS progression. Bile acid synthesis is the predominant pathway for regulating cholesterol in the body [33]. Earlier investigations have highlighted the effect of bile acids on insulin sensitivity, lipid metabolism, and inflammation [34]. Bile acids act as signaling molecules and interact with various receptors, such as the Takeda G protein-coupled receptor 5 (TGR5) and the farnesoid X receptor (FXR) [35]. Activation of these receptors has been shown to influence lipid metabolism, glucose homeostasis, and inflammation, thereby affecting the initiation of AS plaques [36,37,38]. Furthermore, alterations in bile acid composition have been noticed in patients with AS and metabolic syndrome, suggesting a potential connection between bile acid dysregulation and AS [33,39]. Previous studies have reported the association between bile acid levels and age [40]. However, the findings of the present study reveal no significant correlation between elevated bile acid concentrations and age. This is substantiated by the regression coefficient (R^2^) values, indicating that age does not appear to be a determinant factor influencing the bile acid concentrations. Other important metabolites identified in this study were associated with amino acid biosynthesis. Amino acids are not only vital for cellular functions but also contribute to metabolism, lipid regulation, oxidative stress, and inflammation, all of which are key factors in AS [36,41]. Recent studies have indicated that the presence of certain amino acids may contribute to the development of AS [42,43]. In cross-sectional studies, branched-chain amino acids such as leucine, isoleucine, and valine have been associated with a number of cardiometabolic risk factors, including anthropometric measures of excessive body weight and adiposity, insulin resistance, disruption in fasting glucose level, high blood pressure, dyslipidemia, and coronary artery disease [44]. Furthermore, isoleucine impacts the mTOR (mammalian target of rapamycin) signaling pathway, which is involved in metabolism, cell growth, and immune responses [45]. Dysregulated mTOR signaling has been implicated in AS [46]. Another identified metabolite was hypoxanthine, a purine derivative that can be produced due to oxidative stress and impaired purine metabolism [47]. Elevated hypoxanthine levels significantly increased cholesterol levels in serum and the AS plaque area [48]. The presence of plaque in blood vessels can limit blood flow and the amount of oxygen supplied to tissues [3]. This, in turn, affects cellular metabolism and energy production, leading to low ATP (adenosine triphosphate) and high hypoxanthine production [6,49]. It has also been proposed that elevated plasma levels of hypoxanthine in both animal laboratories and clinical settings can be used as rapid and sensitive biomarkers for acute cardiac ischemia at an early stage [6]. Another interesting metabolite found to be significantly upregulated was cortisol, a stress hormone. Cortisol plays a role in the regulation of the immune system, managing lipid and glucose metabolism, and supporting cardiac output through its ability to enhance vascular tone and reduce vascular permeability. Elevated levels of cortisol have several negative health effects throughout the human body. Dysregulation of the hypothalamic-pituitary-adrenal (HPA) axis is linked to hypertension, increased heart rate, high levels of total and low-density lipoprotein cholesterol, as well as fasting insulin and glucose levels [50]. Moreover, several studies have identified an association between cortisol and subclinical atherosclerosis [51]. Lazzarino et al., 2013, found a connection between an increased cortisol response to mental stress and detectable levels of cardiac troponin in plasma using a high-sensitivity test in healthy participants [52].

## 5. Conclusions

In the present study, mass spectrometry-based untargeted and targeted metabolomics combined with univariate and multivariate data analysis was used to identify metabolomics signatures that could discriminate between normal and AS-risk patients. Based on the results, normal and AS-risk sera have significant metabolic differences characterized by changes in several metabolites, including taurocholic acid, cholic acid, cortisol, hypoxanthine, TMAO, isoleucine, glycoursodeoxycholic acid, glycocholic acid, testosterone, leucine, methionine, phenylalanine, tyrosine, and valine. Moreover, ROC analysis and random forest analysis demonstrated that hypoxanthine, isoleucine, and cortisol serum levels are highly correlated with AS, indicating the potential of these metabolites to serve as early biomarkers of AS. These findings provide a new therapeutic direction for developing anti-AS strategies. However, it is recommended to conduct additional longitudinal studies on larger cohorts to confirm the clinical utility and broader applicability of hypoxanthine, isoleucine, and cortisol as early biomarkers for atherosclerosis.

## Figures and Tables

**Figure 1 metabolites-13-01160-f001:**
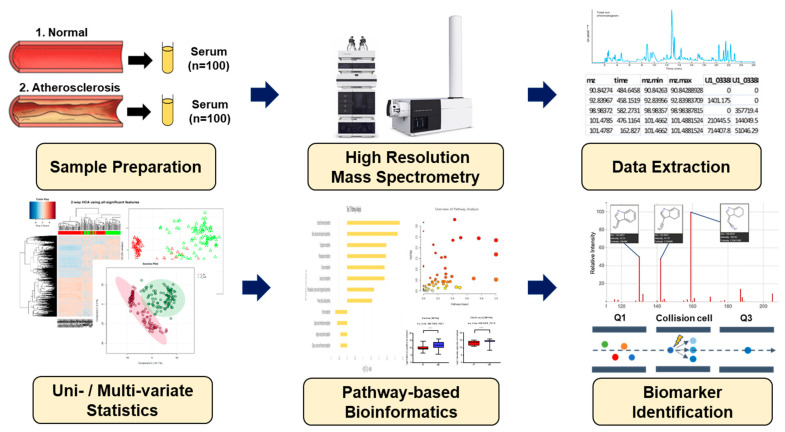
Schematic overview applied in the present study for biomarker identification.

**Figure 2 metabolites-13-01160-f002:**
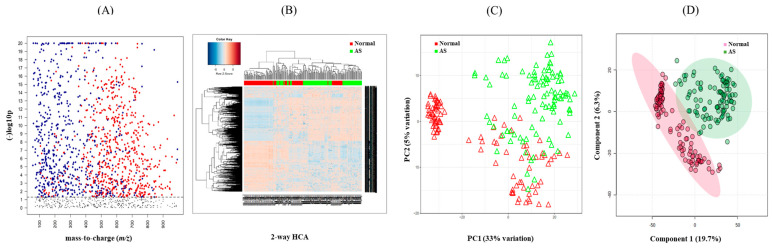
Determination of metabolic signature between normal and AS samples. (**A**) Manhattan plot depicting 1244 significant features (≤0.05 with *t*-test, colored dots) of the two study groups and their distribution along the respective *m*/*z* value. The dotted line showed FDR (q ≤ 0.05), suggesting that features above the line were significantly different between the two groups. (**B**) Hierarchical cluster analysis (HCA) of the samples in the two study groups using 1244 significant features. (**C**) Principal component analysis (PCA) of samples from the two study groups using 1244 significant features. (**D**) PLS-DA of the sera in the two study groups determined 750 features that had the score of variable importance in projection (VIP) ≥ 1.5.

**Figure 3 metabolites-13-01160-f003:**
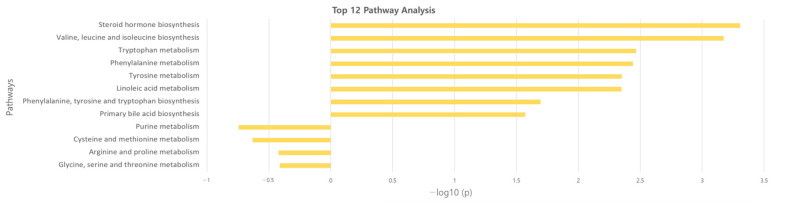
Overview of the disturbed metabolic pathways.

**Figure 4 metabolites-13-01160-f004:**
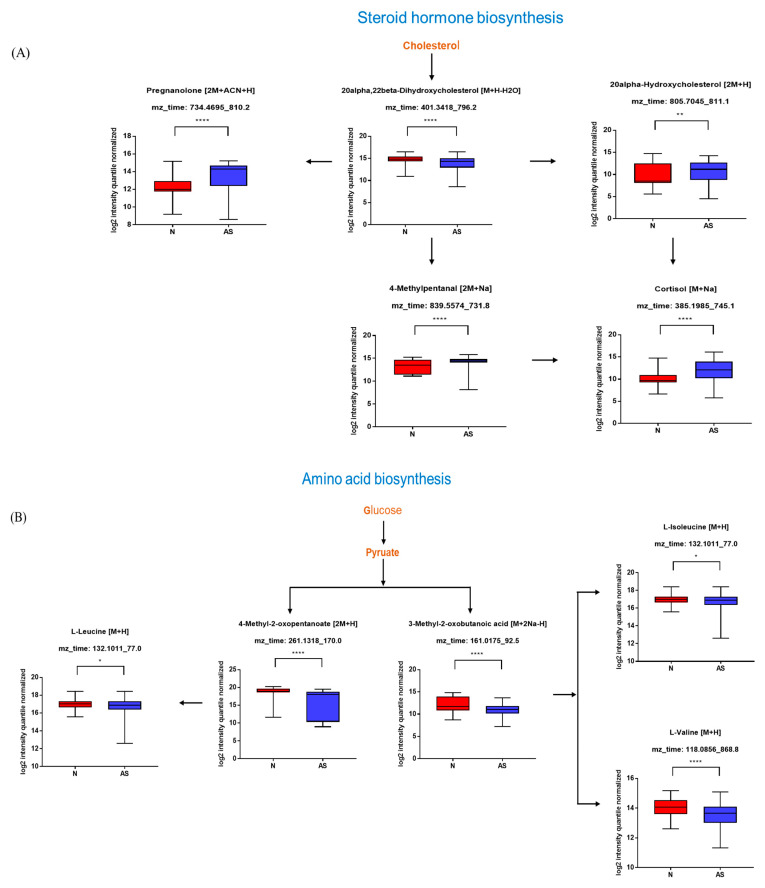
Pathway overview and associated metabolites among normal and AS-risk sera. (**A**) Metabolites associated with steroid hormone biosynthesis. (**B**) Metabolites associated with amino acid biosynthesis. (**C**) Metabolites associated with bile acid biosynthesis. (**D**) Metabolites associated with purine metabolism. * *p* < 0.05; ** *p* < 0.01; *** *p* < 0.005; and **** *p* < 0.001 with Student’s *t*-test.

**Figure 5 metabolites-13-01160-f005:**
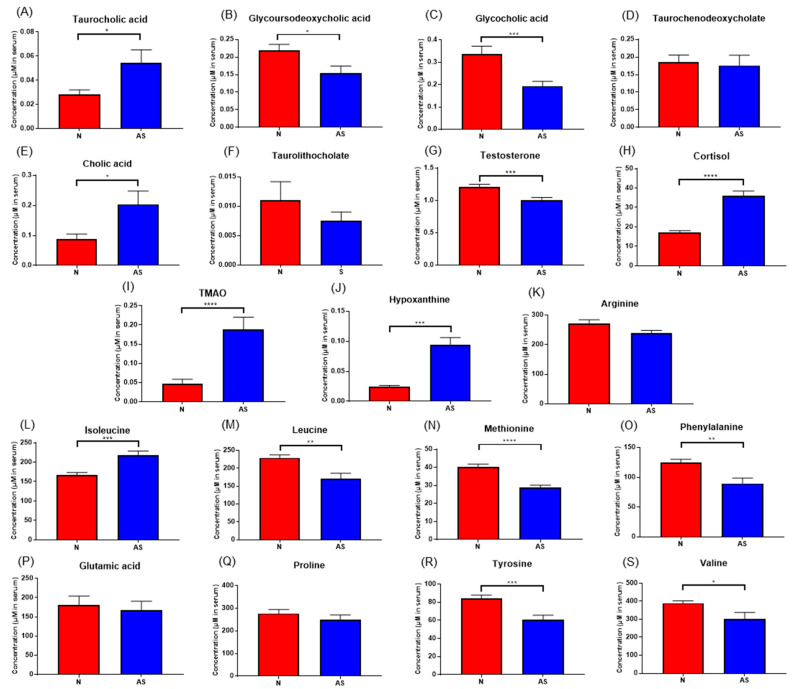
(**A**–**S**) Concentrations of significant metabolites determined via LC-ESI-MS/MS. in serum samples from normal and AS groups. * *p* < 0.05; ** *p* < 0.01; *** *p* < 0.005; and **** *p* < 0.001 with Student’s *t*-test.

**Figure 6 metabolites-13-01160-f006:**
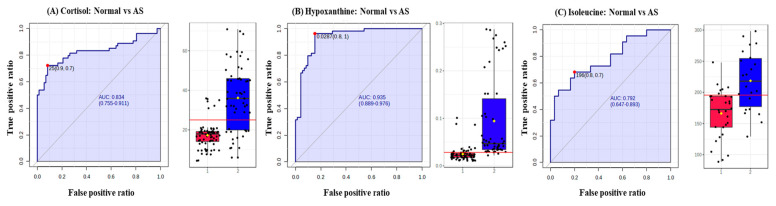
Receiver operating characteristic curves (ROC) and box plots of three potential AS biomarkers. (**A**) ROC curve of cortisol to distinguish between normal and AS groups, (**B**) ROC curve of hypoxanthine to distinguish between normal and AS groups, and ROC curve of isoleucine to distinguish between normal and AS groups. The black dots in the box plots represent the concentration of the selected features from all samples, while the red line in the box plots indicates the optimal cut-point between normal and AS groups.

**Figure 7 metabolites-13-01160-f007:**
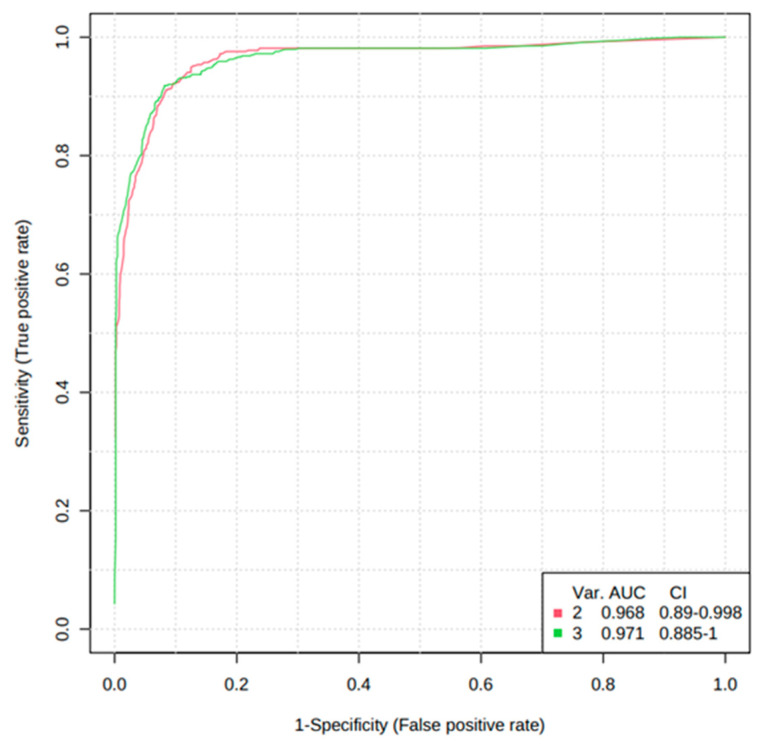
Random forest analysis of the three potential AS biomarkers.

**Table 1 metabolites-13-01160-t001:** Demographic and clinical characteristics of the participants.

	Normal	AS	*p*-Value
Number of patients	100	100	
Gender	Male (100%)	Male (79%), Female (21%)	
Age (year)	19–44	41–101	
Body mass index (kg/m^2^)	25.61 ± 1.799	22.67 ± 0.4179	0.1141
Systolic blood pressure (mmHg)	120.3 ± 0.8022	133.6 ± 2.395	<0.0001
Diastolic blood pressure (mmHg)	76.62 ± 0.6925	73.68 ± 1.185	0.0337
Fasting blood glucose (mg/dL)	92.51 ± 0.6717	128.8 ± 4.71	<0.0001
Total triglyceride (mg/dL)	101.9 ± 2.72	116.7 ± 6.8	0.0449
Total cholesterol (mg/dL)	182.4 ± 2.561	137 ± 4.271	<0.0001
Low-density lipoprotein cholesterol (mg/dL)	107.6 ± 2	79.27 ± 3.292	<0.0001
High-density lipoprotein cholesterol (mg/dL)	52.73 ± 0.973	73.43 ± 4.911	<0.0001

## Data Availability

The data presented in this study are available within the article.

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
