# Peer review of "Identification of Novel Biomarkers for Early Diagnosis of Atherosclerosis Using High-Resolution Metabolomics"

_metabolites, 2023, doi:10.3390/metabo13111160_

Round 1

Reviewer 1 Report

Comments and Suggestions for Authors

 Dear; Editor

 Thank you for providing me with the opportunity to review the manuscript titled "Identification of novel biomarkers for early diagnosis of atherosclerosis using high-resolution metabolomics."

The author has introduced an interesting idea regarding the investigation of bile acid secretions, which has generated significant interest. Bile acids, which are synthesized from cholesterol in the liver, are amphipathic molecules. They play a crucial role in hepatic cholesterol catabolism as a major pathway. Additionally, bile acid synthesis contributes to bile flow, which is essential for the biliary secretion of free cholesterol, endogenous metabolites, and xenobiotics. Bile acids also act as biological detergents that facilitate the absorption of lipids and fat-soluble vitamins in the intestines.

However, I have one comment for the authors regarding the determination and selection of samples. In previous studies, it has been confirmed that there is a close relationship between bile acid metabolism secretion and age. For example, the following articles highlight this association:

- Kiyohisa Uchida, et al., "Age-related changes of bile acid metabolism in rats." Archives of Gerontology and Geriatrics, Volume 10, Issue 1, January–February 1990, Pages 37-48.

- Kurt Einarsson,et al., "Influence of Age on Secretion of Cholesterol and Synthesis of Bile Acids by the Liver." DOI: 10.1056/NEJM198508013130501.

Considering this information, it is important to ensure that the variation in all measured parameters is related to the effectiveness of atherosclerosis (AS) rather than age. Therefore, it is crucial to determine and select patients whose ages are closely matched. 

It is essential for the authors to provide clarification and evidence demonstrating that the variables and discrepancies observed in the results are attributed to the disease itself rather than the substantial age difference between the healthy and diseased samples. 

In page 11, line 335, author write Recent studies have highlighted the effect of bile acids......etc. (ref. no.34) this reference year 2009, that means not recent study, must be delete "recent" from this statement. 

 Reference's lists need to review the format at the beginning of the reference and the sequence of number. 

Best Regards

Reviewer 2 Report

Comments and Suggestions for Authors

In this paper, Sardar et al. investigated the metabolomic profiles of serum from Atherosclerosis patients and healthy individuals using LC-MS/MS. Here are some comments on this study:

1.        In the manuscript, “(p 0.05)” should be p < 0.05. Line 123 “‘d” I assume “rawdata”.

2.        The healthy control group was all male, could the authors give some explanation for this?

3.        In the method part, could the authors provide the criteria for AS enrollment?

4.        Line 153 “(Ogata et al., 1998; Wishart et al., 2018)” has a different citation format.

5.        In Figure 3, is it possible to indicate in which group the increase or decrease occurred?

6.        In Figure 4, the data made a log2 change, which should be stated in the method part.

7.        Is it possible to create a diagnostic model to predict AD based on the metabolite results?

8.        The clarity of the figures needs to be improved.

9.        The reference format needs to be revised, especially the first one.

Reviewer 3 Report

Comments and Suggestions for Authors

 This manuscript  is well written and interesting. Science is good and results have been based on the analysis of a  high number of samples. In my experience this is not very common. I have no particular remarks to make. My only curiosity concerns two parameters indicated in the demographic table (Table 1) i.e., gender and age of subjects. It is not clear to me i) why also the controls did not have the same male/female ratio  as  the patients  and ii) why the authors did not choose controls with an age close to that of patients. Moreover, I  imagine that the number of patients with an age around 40/50 was much lower than that of older subjects. Since the authors well describe other characteristics of the study population, they could probably add a few words to state to underline the homogeneity (or not?) of the two cohorts. This said my opinion is that the manuscript may be accepted with minor revision. 

Round 2

Reviewer 2 Report

Comments and Suggestions for Authors

Thank you for the author's reply. My main points and concerns have been satisfactorily addressed.